# A Statistical Decision-Theoretic Framework for Social Choice

Hossein Azari Soufiani[*]        David C. Parkes [†]        Lirong Xia[‡]

## Abstract

In this paper, we take a statistical decision-theoretic viewpoint on social choice, putting a focus on the decision to be made on behalf of a system of agents. In our framework, we are given a statistical ranking model, a decision space, and a loss function defined on (parameter, decision) pairs, and formulate social choice mechanisms as decision rules that minimize expected loss. This suggests a general framework for the design and analysis of new social choice mechanisms. We compare *Bayesian estimators*, which minimize Bayesian expected loss, for the Mallows model and the Condorcet model respectively, and the Kemeny rule. We consider various normative properties, in addition to computational complexity and asymptotic behavior. In particular, we show that the Bayesian estimator for the Condorcet model satisfies some desired properties such as anonymity, neutrality, and monotonicity, can be computed in polynomial time, and is asymptotically different from the other two rules when the data are generated from the Condorcet model for some ground truth parameter.

## 1  Introduction

Social choice studies the design and evaluation of voting rules (or rank aggregation rules). There have been two main perspectives: reach a compromise among subjective preferences of agents, or make an objectively correct decision. The former has been extensively studied in classical social choice in the context of political elections, while the latter is relatively less developed, even though it can be dated back to the Condorcet Jury Theorem in the 18th century [9].

In many multi-agent and social choice scenarios the main consideration is to achieve the second objective, and make an objectively correct decision. Meanwhile, we also want to respect agents' preferences and opinions, and require the voting rule to satisfy well-established normative properties in social choice. For example, when a group of friends vote to choose a restaurant for dinner, perhaps the most important goal is to find an objectively good restaurant, but it is also important to use a good voting rule in the social choice sense. Even for applications with less societal context, e.g. using voting rules to aggregate rankings in meta-search engines [12], recommender systems [15], crowdsourcing [23], semantic webs [27], some social choice normative properties are still desired. For example, *monotonicity* may be desired, which requires that raising the position of an alternative in any vote does not hurt the alternative in the outcome of the voting rule. In addition, we require voting rules to be efficiently computable.

Such scenarios propose the following new challenge: *How can we design new voting rules with good statistical properties as well as social choice normative properties?*

To tackle this challenge, we develop a general framework that adopts *statistical decision theory* [3]. Our approach couples a statistical ranking model with an explicit decision space and loss function.

---

[*]azari@google.com, Google Research, New York, NY 10011, USA. The work was done when the author was at Harvard University.

[†]parkes@eecs.harvard.edu, Harvard University, Cambridge, MA 02138, USA.

[‡]xial@cs.rpi.edu, Rensselaer Polytechnic Institute, Troy, NY 12180, USA.

| | Anonymity, neutrality Monotonicity | Majority, Condorcet | Consistency | Complexity | Min. Bayesian risk |
|---|---|---|---|---|---|
| Kemeny | Y | Y | N | NP-hard, $P_{||}^{NP}$-hard | N |
| Bayesian est. of $\mathcal{M}_\varphi^1$ (uni. prior) | Y | N | N | NP-hard, $P_{||}^{NP}$-hard (Theorem 3) | Y |
| Bayesian est. of $\mathcal{M}_\varphi^2$ (uni. prior) | Y | N | N | P (Theorem 4) | Y |

Table 1: Kemeny for winners vs. Bayesian estimators of $\mathcal{M}_\varphi^1$ and $\mathcal{M}_\varphi^2$ to choose *winners*.

Given these, we can adopt *Bayesian estimators* as social choice mechanisms, which make decisions to minimize the expected loss w.r.t. the posterior distribution on the parameters (called the *Bayesian risk*). This provides a principled methodology for the design and analysis of new voting rules.

To show the viability of the framework, we focus on selecting multiple alternatives (the alternatives that can be thought of as being "tied" for the first place) under a natural extension of the 0-1 loss function for two models: let $\mathcal{M}_\varphi^1$ denote the *Mallows model* with fixed dispersion [22], and let $\mathcal{M}_\varphi^2$ denote the *Condorcet model* proposed by Condorcet in the 18th century [9, 34]. In both models the dispersion parameter, denoted $\varphi$, is taken as a fixed parameter. The difference is that in the Mallows model the parameter space is composed of all linear orders over alternatives, while in the Condorcet model the parameter space is composed of all possibly cyclic rankings over alternatives (irreflexive, antisymmetric, and total binary relations). $\mathcal{M}_\varphi^2$ is a natural model that captures real-world scenarios where the ground truth may contain cycles, or agents' preferences are cyclic, but they have to report a linear order due to the protocol. More importantly, as we will show later, a Bayesian estimator on $\mathcal{M}_\varphi^2$ is superior from a computational viewpoint.

Through this approach, we obtain two voting rules as Bayesian estimators and then evaluate them with respect to various normative properties, including *anonymity, neutrality, monotonicity, the majority criterion, the Condorcet criterion* and *consistency*. Both rules satisfy anonymity, neutrality, and monotonicity, but fail the majority criterion, Condorcet criterion,[1] and consistency. Admittedly, the two rules do not enjoy outstanding normative properties, but they are not bad either. We also investigate the computational complexity of the two rules. Strikingly, despite the similarity of the two models, the Bayesian estimator for $\mathcal{M}_\varphi^2$ can be computed in polynomial time, while computing the Bayesian estimator for $\mathcal{M}_\varphi^1$ is $P_{||}^{NP}$-hard, which means that it is at least NP-hard. Our results are summarized in Table 1.

We also compare the asymptotic outcomes of the two rules with the Kemeny rule for winners, which is a natural extension of the maximum likelihood estimator of $\mathcal{M}_\varphi^1$ proposed by Fishburn [14]. It turns out that when $n$ votes are generated under $\mathcal{M}_\varphi^1$, all three rules select the same winner asymptotically almost surely (a.a.s.) as $n \to \infty$. When the votes are generated according to $\mathcal{M}_\varphi^2$, the rule for $\mathcal{M}_\varphi^1$ still selects the same winner as Kemeny a.a.s.; however, for some parameters, the winner selected by the rule for $\mathcal{M}_\varphi^2$ is different with non-negligible probability. These are confirmed by experiments on synthetic datasets.

**Related work.** Along the second perspective in social choice (to make an objectively correct decision), in addition to Condorcet's statistical approach to social choice [9, 34], most previous work in economics, political science, and statistics focused on extending the theorem to heterogeneous, correlated, or strategic agents for two alternatives, see [25, 1] among many others. Recent work in computer science views agents' votes as i.i.d. samples from a statistical model, and computes the MLE to estimate the parameters that maximize the likelihood [10, 11, 33, 32, 2, 29, 7]. A limitation of these approaches is that they estimate the parameters of the model, but may not directly inform the right *decision* to make in the multi-agent context. The main approach has been to return the modal rank order implied by the estimated parameters, or the alternative with the highest, predicted marginal probability of being ranked in the top position.

There have also been some proposals to go beyond MLE in social choice. In fact, Young [34] proposed to select a winning alternative that is *"most likely to be the best (i.e., top-ranked in the true ranking)"* and provided formulas to compute it for three alternatives. This idea has been formalized and extended by Procaccia et al. [29] to choose a given number of alternatives with highest marginal

probability under the Mallows model. More recently, independent to our work, Elkind and Shah [13] investigated a similar question for choosing multiple winners under the Condorcet model. We will see that these are special cases of our proposed framework in Example 2. Pivato [26] conducted a similar study to Conitzer and Sandholm [10], examining voting rules that can be interpreted as expect-utility maximizers.

We are not aware of previous work that frames the problem of social choice from the viewpoint of statistical decision theory, which is our main conceptual contribution. Technically, the approach taken in this paper advocates a general paradigm of "design by statistics, evaluation by social choice and computer science". We are not aware of a previous work following this paradigm to design and evaluate new rules. Moreover, the normative properties for the two voting rules investigated in this paper are novel, even though these rules are not really novel. Our result on the computational complexity of the first rule strengthens the NP-hardness result by Procaccia et al. [29], and the complexity for the second rule (Theorem 5) was independently discovered by Elkind and Shah [13].

The statistical decision-theoretic framework is quite general, allowing considerations such as estimators that minimize the maximum expected loss, or the maximum expected regret [3]. In a different context, focused on uncertainty about the availability of alternatives, Lu and Boutilier [20] adopt a decision-theoretic view of the design of an optimal voting rule. Caragiannis et al. [8] studied the robustness of social choice mechanisms w.r.t. model uncertainty, and characterized a unique social choice mechanism that is consistent w.r.t. a large class of ranking models.

A number of recent papers in computational social choice take utilitarian and decision-theoretical approaches towards social choice [28, 6, 4, 5]. Most of them evaluate the joint decision w.r.t. agents' *subjective* preferences, for example the sum of agents' subjective utilities (i.e. the *social welfare*). We don't view this as fitting into the classical approach to *statistical* decision theory as formulated by Wald [30]. In our framework, the joint decision is evaluated objectively w.r.t. the ground truth in the statistical model. Several papers in machine learning developed algorithms to compute MLE or Bayesian estimators for popular ranking models [18, 19, 21], but without considering the normative properties of the estimators.

## 2 Preliminaries

In social choice, we have a set of $m$ alternatives $\mathcal{C} = \{c_1, \ldots, c_m\}$ and a set of $n$ agents. Let $\mathcal{L}(\mathcal{C})$ denote the set of all linear orders over $\mathcal{C}$. For any alternative $c$, let $\mathcal{L}_c(\mathcal{C})$ denote the set of linear orders over $\mathcal{C}$ where $c$ is ranked at the top. Agent $j$ uses a linear order $V_j \in \mathcal{L}(\mathcal{C})$ to represent her preferences, called her *vote*. The collection of agents votes is called a *profile*, denoted by $P = \{V_1, \ldots, V_n\}$. A *(irresolute) voting rule* $r : \mathcal{L}(\mathcal{C})^n \to (2^{\mathcal{C}} \setminus \emptyset)$ selects a set of winners that are "tied" for the first place for every profile of $n$ votes.

For any pair of linear orders $V, W$, let $\text{Kendall}(V, W)$ denote the *Kendall-tau distance* between $V$ and $W$, that is, the number of different pairwise comparisons in $V$ and $W$. The *Kemeny rule* (a.k.a. *Kemeny-Young method*) [17, 35] selects all *linear orders* with the minimum Kendall-tau distance from the preference profile $P$, that is, $\text{Kemeny}(P) = \arg\min_W \text{Kendall}(P, W)$. The most well-known variant of Kemeny to select winning alternatives, denoted by $\text{Kemeny}_{\mathcal{C}}$, is due to Fishburn [14], who defined it as a voting rule that selects all alternatives that are ranked in the top position of some winning linear orders under the Kemeny rule. That is, $\text{Kemeny}_{\mathcal{C}}(P) = \{top(V) : V \in \text{Kemeny}(P)\}$, where $top(V)$ is the top-ranked alternative in $V$.

Voting rules are often evaluated by the following normative properties. An irresolute rule $r$ satisfies:

- *anonymity*, if $r$ is insensitive to permutations over agents;
- *neutrality*, if $r$ is insensitive to permutations over alternatives;
- *monotonicity*, if for any $P$, $c \in r(P)$, and any $P'$ that is obtained from $P$ by only raising the positions of $c$ in one or multiple votes, then $c \in r(P')$;
- *Condorcet criterion*, if for any profile $P$ where a Condorcet winner exists, it must be the unique winner. A Condorcet winner is the alternative that beats every other alternative in pair-wise elections.
- *majority criterion*, if for any profile $P$ where an alternative $c$ is ranked in the top positions for more than half of the votes, then $r(P) = \{c\}$. If $r$ satisfies Condorcet criterion then it also satisfies the majority criterion.
- *consistency*, if for any pair of profiles $P_1, P_2$ with $r(P_1) \cap r(P_2) \neq \emptyset$, $r(P_1 \cup P_2) = r(P_1) \cap r(P_2)$.

For any profile $P$, its *weighted majority graph (WMG)*, denoted by $\text{WMG}(P)$, is a weighted directed graph whose vertices are $\mathcal{C}$, and there is an edge between any pair of alternatives $(a, b)$ with weight $w_P(a, b) = \#\{V \in P : a \succ_V b\} - \#\{V \in P : b \succ_V a\}$.

A parametric model $\mathcal{M} = (\Theta, \mathcal{S}, \text{Pr})$ is composed of three parts: a *parameter space* $\Theta$, a *sample space* $\mathcal{S}$ composing of all datasets, and a set of probability distributions over $\mathcal{S}$ indexed by elements of $\Theta$: for each $\theta \in \Theta$, the distribution indexed by $\theta$ is denoted by $\text{Pr}(\cdot|\theta)$.[2]

Given a parametric model $\mathcal{M}$, a *maximum likelihood estimator (MLE)* is a function $f_{\text{MLE}} : \mathcal{S} \to \Theta$ such that for any data $P \in \mathcal{S}$, $f_{\text{MLE}}(P)$ is a parameter that maximizes the likelihood of the data. That is, $f_{\text{MLE}}(P) \in \arg\max_{\theta \in \Theta} \text{Pr}(P|\theta)$.

In this paper we focus on *parametric ranking models*. Given $\mathcal{C}$, a parametric ranking model $\mathcal{M}_{\mathcal{C}} = (\Theta, \text{Pr})$ is composed of a parameter space $\Theta$ and a distribution $\text{Pr}(\cdot|\theta)$ over $\mathcal{L}(\mathcal{C})$ for each $\theta \in \Theta$, such that for any number of voters $n$, the sample space is $\mathcal{S}_n = \mathcal{L}(\mathcal{C})^n$, where each vote is generated i.i.d. from $\text{Pr}(\cdot|\theta)$. Hence, for any profile $P \in \mathcal{S}_n$ and any $\theta \in \Theta$, we have $\text{Pr}(P|\theta) = \prod_{V \in P} \text{Pr}(V|\theta)$. We omit the sample space because it is determined by $\mathcal{C}$ and $n$.

**Definition 1** *In the* Mallows model *[22], a parameter is composed of a linear order $W \in \mathcal{L}(\mathcal{C})$ and a* dispersion *parameter $\varphi$ with $0 < \varphi < 1$. For any profile $P$ and $\theta = (W, \varphi)$, $\text{Pr}(P|\theta) = \prod_{V \in P} \frac{1}{Z} \varphi^{Kendall(V,W)}$, where $Z$ is the normalization factor with $Z = \sum_{V \in \mathcal{L}(\mathcal{C})} \varphi^{Kendall(V,W)}$.*

*Statistical decision theory* [30, 3] studies scenarios where the decision maker must make a *decision* $d \in \mathcal{D}$ based on the data $P$ generated from a parametric model, generally $\mathcal{M} = (\Theta, \mathcal{S}, \text{Pr})$. The quality of the decision is evaluated by a *loss function* $L : \Theta \times \mathcal{D} \to \mathbb{R}$, which takes the *true* parameter and the decision as inputs.

In this paper, we focus on the *Bayesian principle* of statistical decision theory to design social choice mechanisms as choice functions that minimize the *Bayesian risk* under a prior distribution over $\Theta$. More precisely, the Bayesian risk, $R_B(P, d)$, is the expected loss of the decision $d$ when the parameter is generated according to the posterior distribution given data $P$. That is, $R_B(P, d) = E_{\theta|P} L(\theta, d)$. Given a parametric model $\mathcal{M}$, a loss function $L$, and a prior distribution over $\Theta$, a (deterministic) *Bayesian estimator* $f_B$ is a decision rule that makes a deterministic decision in $\mathcal{D}$ to minimize the Bayesian risk, that is, for any $P \in \mathcal{S}$, $f_B(P) \in \arg\min_d R_B(P, d)$. We focus on deterministic estimators in this work and leave randomized estimators for future research.

**Example 1** *When $\Theta$ is discrete, an MLE of a parametric model $\mathcal{M}$ is a Bayesian estimator of the statistical decision problem $(\mathcal{M}, \mathcal{D} = \Theta, L_{0\text{-}1})$ under the uniform prior distribution, where $L_{0\text{-}1}$ is the 0-1 loss function such that $L_{0\text{-}1}(\theta, d) = 0$ if $\theta = d$, otherwise $L_{0\text{-}1}(\theta, d) = 1$.*

In this sense, all previous MLE approaches in social choice can be viewed as the Bayesian estimators of a statistical decision-theoretic framework for social choice where $\mathcal{D} = \Theta$, a 0-1 loss function, and the uniform prior.

## 3 Our Framework

Our framework is quite general and flexible because we can choose any parametric ranking model, any decision space, any loss function, and any prior to use the Bayesian estimators social choice mechanisms. Common choices of both $\Theta$ and $\mathcal{D}$ are $\mathcal{L}(\mathcal{C})$, $\mathcal{C}$, and $(2^{\mathcal{C}} \setminus \emptyset)$.

**Definition 2** *A* statistical decision-theoretic framework for social choice *is a tuple $\mathcal{F} = (\mathcal{M}_{\mathcal{C}}, \mathcal{D}, L)$, where $\mathcal{C}$ is the set of alternatives, $\mathcal{M}_{\mathcal{C}} = (\Theta, \text{Pr})$ is a parametric ranking model, $\mathcal{D}$ is the decision space, and $L : \Theta \times \mathcal{D} \to \mathbb{R}$ is a loss function.*

Let $\mathcal{B}(\mathcal{C})$ denote the set of all irreflexive, antisymmetric, and total binary relations over $\mathcal{C}$. For any $c \in \mathcal{C}$, let $\mathcal{B}_c(\mathcal{C})$ denote the relations in $\mathcal{B}(\mathcal{C})$ where $c \succ a$ for all $a \in \mathcal{C} - \{c\}$. It follows that $\mathcal{L}(\mathcal{C}) \subseteq \mathcal{B}(\mathcal{C})$, and moreover, the Kendall-tau distance can be defined to count the number of pairwise disagreements between elements of $\mathcal{B}(\mathcal{C})$.

In the rest of the paper, we focus on the following two parametric ranking models, where the dispersion is a fixed parameter.

**Definition 3 (Mallows model with fixed dispersion, and the Condorcet model)** *Let $\mathcal{M}_\varphi^1$ denote the* Mallows model with fixed dispersion, *where the parameter space is $\Theta = \mathcal{L}(\mathcal{C})$ and given any $W \in \Theta$, $\Pr(\cdot|W)$ is $\Pr(\cdot|(W, \varphi))$ in the Mallows model, where $\varphi$ is fixed.*

*In the Condorcet model, $\mathcal{M}_\varphi^2$, the parameter space is $\Theta = \mathcal{B}(\mathcal{C})$. For any $W \in \Theta$ and any profile $P$, we have $\Pr(P|W) = \prod_{V \in P} \left(\frac{1}{Z} \varphi^{Kendall(V,W)}\right)$, where $Z$ is the normalization factor such that $Z = \sum_{V \in \mathcal{B}(\mathcal{C})} \varphi^{Kendall(V,W)}$, and parameter $\varphi$ is fixed.*[3]

$\mathcal{M}_\varphi^1$ and $\mathcal{M}_\varphi^2$ degenerate to the Condorcet model for two alternatives [9]. The Kemeny rule that selects a linear order is an MLE of $\mathcal{M}_\varphi^1$ for any $\varphi$.

We now formally define two statistical decision-theoretic frameworks associated with $\mathcal{M}_\varphi^1$ and $\mathcal{M}_\varphi^2$, which are the focus of the rest of our paper.

**Definition 4** *For $\Theta = \mathcal{L}(\mathcal{C})$ or $\mathcal{B}(\mathcal{C})$, any $\theta \in \Theta$, and any $c \in \mathcal{C}$, we define a loss function $L_{top}(\theta, c)$ such that $L_{top}(\theta, c) = 0$ if for all $b \in \mathcal{C}$, $c \succ b$ in $\theta$; otherwise $L_{top}(\theta, c) = 1$.*

*Let $\mathcal{F}_\varphi^1 = (\mathcal{M}_\varphi^1, 2^{\mathcal{C}} \setminus \emptyset, L_{top})$ and $\mathcal{F}_\varphi^2 = (\mathcal{M}_\varphi^2, 2^{\mathcal{C}} \setminus \emptyset, L_{top})$, where for any $C \subseteq \mathcal{C}$, $L_{top}(\theta, C) = \sum_{c \in C} L_{top}(\theta, c)/|C|$. Let $f_B^1$ (respectively, $f_B^2$) denote the Bayesian estimators of $\mathcal{F}_\varphi^1$ (respectively, $\mathcal{F}_\varphi^2$) under the uniform prior.*

We note that $L_{top}$ in the above definition takes a parameter and a decision in $2^{\mathcal{C}} \setminus \emptyset$ as inputs, which makes it different from the 0-1 loss function $L_{0\text{-}1}$ that takes a pair of parameters as inputs, as the one in Example 1. Hence, $f_B^1$ and $f_B^2$ are *not* the MLEs of their respective models, as was the case in Example 1. We focus on voting rules obtained by our framework with $L_{top}$. Certainly our framework is not limited to this loss function.

**Example 2** *Bayesian estimators $f_B^1$ and $f_B^2$ coincide with Young [34]'s idea of selecting the alternative that is "most likely to be the best (i.e., top-ranked in the true ranking)", under $\mathcal{F}_\varphi^1$ and $\mathcal{F}_\varphi^2$ respectively. This gives a theoretical justification of Young's idea and other followups under our framework. Specifically, $f_B^1$ is similar to rule studied by Procaccia et al. [29] and $f_B^2$ was independently studied by Elkind and Shah [13].*

## 4 Normative Properties of Bayesian Estimators

All omitted proofs can be found in the full version on arXiv.

**Theorem 1** *For any $\varphi$, $f_B^1$ satisfies anonymity, neutrality, and monotonicity. $f_B^1$ does not satisfy majority or the Condorcet criterion for any $\varphi < \frac{1}{\sqrt{2}}$,*[4] *and it does not satisfy consistency.*

**Proof sketch:** Anonymity and neutrality are obviously satisfied.

**Monotonicity.** Monotonicity follows from the following lemma.

**Lemma 1** *For any $c \in \mathcal{C}$, let $P'$ denote a profile obtained from $P$ by raising the position of $c$ in one vote. For any $W \in \mathcal{L}_c(\mathcal{C})$, $Pr(P'|W) = Pr(P|W)/\varphi$; for any $b \in \mathcal{C}$ and any $V \in \mathcal{L}_b(\mathcal{C})$, $Pr(P'|V) \leq Pr(P|V)/\varphi$.*

**Majority and the Condorcet criterion.** Let $\mathcal{C} = \{c, b, c_3, \ldots, c_m\}$. We construct a profile $P^*$ where $c$ is ranked in the top positions for more than half of the votes, but $c \notin f_B^1(P^*)$.

For any $k$, let $P^*$ denote a profile composed of $k$ copies of $[c \succ b \succ c_3 \succ \cdots \succ c_m]$, 1 of $[c \succ b \succ c_m \succ \cdots \succ c_3]$ and $k-1$ copies of $[b \succ c_m \succ \cdots \succ c_3 \succ c]$. It is not hard to verify that the WMG of $P^*$ is as in Figure 1 (a).

Then, we prove that for any $\varphi < \frac{1}{\sqrt{2}}$, we can find $m$ and $k$ so that $\frac{\sum_{V \in \mathcal{L}_c(\mathcal{C})} \Pr(P|V)}{\sum_{W \in \mathcal{L}_b(\mathcal{C})} \Pr(P|W)} = \frac{1 + \varphi^{2k} + \cdots + \varphi^{2k(m-2)}}{1 + \varphi^2 + \cdots + \varphi^{2(m-2)}} \cdot \varphi^2 < 1$. It follows that $c$ is the Condorcet winner in $P^*$ but it does not minimize the Bayesian risk under $\mathcal{M}_\varphi^1$, which means that it is not the winner under $f_B^1$.

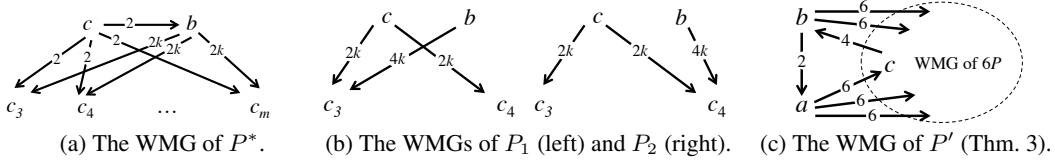

(a) The WMG of $P^*$.          (b) The WMGs of $P_1$ (left) and $P_2$ (right).          (c) The WMG of $P'$ (Thm. 3).

Figure 1: WMGs of the profiles for proofs: (a) for majority and Condorcet (Thm. 1); (b) for consistency (Thm. 1); (c) for computational complexity (Thm. 3).

**Consistency.** We construct an example to show that $f_B^1$ does not satisfy consistency. In our construction $m$ and $n$ are even, and $\mathcal{C} = \{c, b, c_3, c_4\}$. Let $P_1$ and $P_2$ denote profiles whose WMGs are as shown in Figure 1 (b), respectively. We have the following lemma.

**Lemma 2** *Let $P \in \{P_1, P_2\}$, $\frac{\sum_{V \in \mathcal{L}_c(\mathcal{C})} \Pr(P|V)}{\sum_{W \in \mathcal{L}_b(\mathcal{C})} \Pr(P|W)} = \frac{3(1+\varphi^{4k})}{2(1+\varphi^{2k}+\varphi^{4k})}$.*

For any $0 < \varphi < 1$, $\frac{3(1+\varphi^{4k})}{2(1+\varphi^{2k}+\varphi^{4k})} > 1$ for all $k$. It is not hard to verify that $f_B^1(P_1) = f_B^1(P_2) = \{c\}$ and $f_B^1(P_1 \cup P_2) = \{c, b\}$, which means that $f_B^1$ is not consistent. $\qquad\square$

Similarly, we can prove the following theorem for $f_B^2$.

**Theorem 2** *For any $\varphi$, $f_B^2$ satisfies anonymity, neutrality, and monotonicity. It does not satisfy majority, the Condorcet criterion, or consistency.*

By Theorem 1 and 2, $f_B^1$ and $f_B^2$ do not satisfy as many desired normative properties as the Kemeny rule for winners. On the other hand, they minimize Bayesian risk under $\mathcal{F}_\varphi^1$ and $\mathcal{F}_\varphi^2$, respectively, for which Kemeny does neither. In addition, neither $f_B^1$ nor $f_B^2$ satisfy consistency, which means that they are not positional scoring rules.

# 5 Computational Complexity

We consider the following two types of decision problems.

**Definition 5** *In the* BETTER BAYESIAN DECISION *problem for a statistical decision-theoretic framework $(\mathcal{M}_\mathcal{C}, \mathcal{D}, L)$ under a prior distribution, we are given $d_1, d_2 \in \mathcal{D}$, and a profile $P$. We are asked whether $R_B(P, d_1) \le R_B(P, d_2)$.*

We are also interested in checking whether a given alternative is the optimal decision.

**Definition 6** *In the* OPTIMAL BAYESIAN DECISION *problem for a statistical decision-theoretic framework $(\mathcal{M}_\mathcal{C}, \mathcal{D}, L)$ under a prior distribution, we are given $d \in \mathcal{D}$ and a profile $P$. We are asked whether $d$ minimizes the Bayesian risk $R_B(P, \cdot)$.*

$\mathsf{P}_{||}^{\mathsf{NP}}$ is the class of decision problems that can be computed by a $P$ oracle machine with polynomial number of parallel calls to an $\mathsf{NP}$ oracle. A decision problem $A$ is $\mathsf{P}_{||}^{\mathsf{NP}}$-hard, if for any $\mathsf{P}_{||}^{\mathsf{NP}}$ problem $B$, there exists a polynomial-time many-one reduction from $B$ to $A$. It is known that $\mathsf{P}_{||}^{\mathsf{NP}}$-hard problems are $\mathsf{NP}$-hard.

**Theorem 3** *For any $\varphi$,* BETTER BAYESIAN DECISION *and* OPTIMAL BAYESIAN DECISION *for $\mathcal{F}_\varphi^1$ under uniform prior are $\mathsf{P}_{||}^{\mathsf{NP}}$-hard.*

**Proof:** The hardness of both problems is proved by a unified reduction from the KEMENY WINNER problem, which is $\mathsf{P}_{||}^{\mathsf{NP}}$-complete [16]. In a KEMENY WINNER problem, we are given a profile $P$ and an alternative $c$, and we are asked if $c$ is ranked in the top of at least one $V \in \mathcal{L}(\mathcal{C})$ that minimizes $\text{Kendall}(P, V)$.

For any alternative $c$, the *Kemeny score* of $c$ under $\mathcal{M}_\varphi^1$ is the smallest distance between the profile $P$ and any linear order where $c$ is ranked in the top. We next prove that when $\varphi < \frac{1}{m!}$, the Bayesian risk of $c$ is largely determined by the Kemeny score of $c$.

**Lemma 3** *For any $\varphi < \frac{1}{m!}$, any $c, b \in \mathcal{C}$, and any profile $P$, if the Kemeny score of $c$ is strictly smaller than the Kemeny score of $b$ in $P$, then $R_B(P, c) < R_B(P, b)$ for $\mathcal{M}_\varphi^1$.*

Let $t$ be any natural number such that $\varphi^t < \frac{1}{m!}$. For any KEMENY WINNER instance $(P, c)$ for alternatives $\mathcal{C}'$, we add two more alternatives $\{a, b\}$ and define a profile $P'$ whose WMG is as shown in Figure 3(c) using McGarvey's trick [24]. The WMG of $P'$ contains the WMG$(P)$ as a subgraph, where the weights are 6 times the weights in WMG$(P)$.

Then, we let $P^* = tP'$, which is $t$ copies of $P'$. It follows that for any $V \in \mathcal{L}(\mathcal{C})$, $\Pr(P^*|V, \varphi) = \Pr(P'|V, \varphi^t)$. By Lemma 3, if an alternative $e$ has the strictly lowest Kemeny score for profile $P'$, then it the unique alternative that minimizes the Bayesian risk for $P'$ and dispersion parameter $\varphi^t$, which means that $e$ minimizes the Bayesian risk for $P^*$ and dispersion parameter $\varphi$.

Let $O$ denote the set of linear orders over $\mathcal{C}'$ that minimizes the Kendall tau distance from $P$ and let $k$ denote this minimum distance. Choose an arbitrary $V' \in O$. Let $V = [b \succ a \succ V']$. It follows that Kendall$(P', V) = 4 + 6k$. If there exists $W' \in O$ where $c$ is ranked in the top position, then we let $W = [a \succ c \succ b \succ (V' - \{c\})]$. We have Kendall$(P', W) = 2 + 6k$. If $c$ is not a Kemeny winner in $P$, then for any $W$ where $d$ is not ranked in the top position, Kendall$(P', W) \geq 6 + 6k$. Therefore, $a$ minimizes the Bayesian risk if and only if $c$ is a Kemeny winner in $P$, and if $c$ does not minimize the Bayesian risk, then $b$ does. Hence BETTER DECISION (checking if $a$ is better than $b$) and OPTIMAL BAYESIAN DECISION (checking if $a$ is the optimal alternative) are $\mathsf{P}_{||}^{\mathsf{NP}}$-hard. $\quad\square$

We note that OPTIMAL BAYESIAN DECISION in Theorem 3 is equivalent to checking whether a given alternative $c$ is in $f_B^1(P)$. We do not know whether these problems are $\mathsf{P}_{||}^{\mathsf{NP}}$-complete. In sharp contrast to $f_B^1$, the next theorem states that $f_B^2$ under uniform prior is in P.

**Theorem 4** *For any rational number[5] $\varphi$,* BETTER BAYESIAN DECISION *and* OPTIMAL BAYESIAN DECISION *for $\mathcal{F}_\varphi^2$ under uniform prior are in* P.

The theorem is a corollary of the following stronger theorem that provides a closed-form formula for Bayesian loss for $\mathcal{F}_\varphi^2$.[6] We recall that for any profile $P$ and any pair of alternatives $c, b$, that $w_P(c, b)$ is the weight on $c \to b$ in the weighted majority graph of $P$.

**Theorem 5** *For $\mathcal{F}_\varphi^2$ under uniform prior, for any $c \in \mathcal{C}$ and any profile $P$, $R_B(P, c) = 1 - \prod_{b \neq c} \dfrac{1}{1 + \varphi^{w_P(c,b)}}$.*

The comparisons of Kemeny, $f_B^1$, and $f_B^2$ are summarized in Table 1. According to the criteria we consider, none of the three outperforms the others. Kemeny does well in normative properties, but does not minimize Bayesian risk under either $\mathcal{F}_\varphi^1$ or $\mathcal{F}_\varphi^2$, and is hard to compute. $f_B^1$ minimizes the Bayesian risk under $\mathcal{F}_\varphi^1$, but is hard to compute. We would like to highlight $f_B^2$, which minimizes the Bayesian risk under $\mathcal{F}_\varphi^2$, and more importantly, can be computed in polynomial time despite the similarity between $\mathcal{F}_\varphi^1$ and $\mathcal{F}_\varphi^2$.

## 6 Asymptotic Comparisons

In this section, we ask the following question: as the number of voters, $n \to \infty$, what is the probability that Kemeny, $f_B^1$, and $f_B^2$ choose different winners? We show that when the data is generated from $\mathcal{M}_\varphi^1$, all three methods are equal *asymptotically almost surely (a.a.s.)*, that is, they are equal with probability 1 as $n \to \infty$.

**Theorem 6** *Let $P_n$ denote a profile of $n$ votes generated i.i.d. from $\mathcal{M}_\varphi^1$ given $W \in \mathcal{L}_c(\mathcal{C})$. Then, $\Pr_{n \to \infty}(Kemeny(P_n) = f_B^1(P_n) = f_B^2(P_n) = c) = 1$.*

However, when the data are generated from $\mathcal{M}_\varphi^2$, we have a different story.

**Theorem 7** *For any $W \in \mathcal{B}(\mathcal{C})$ and any $\varphi$, $f_B^1(P_n) = Kemeny(P_n)$ a.a.s. as $n \to \infty$ and votes in $P_n$ are generated i.i.d. from $\mathcal{M}_\varphi^2$ given $W$.*

*For any $m \geq 5$, there exists $W \in \mathcal{B}(\mathcal{C})$ such that for any $\varphi$, there exists $\epsilon > 0$ such that with probability at least $\epsilon$, $f_B^1(P_n) \neq f_B^2(P_n)$ and Kemeny$(P_n) \neq f_B^2(P_n)$ as $n \to \infty$ and votes in $P_n$ are generated i.i.d. from $\mathcal{M}_\varphi^2$ given $W$.*

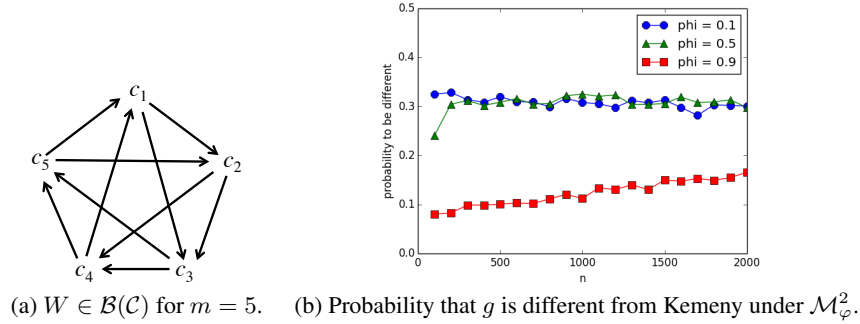

(a) $W \in \mathcal{B}(\mathcal{C})$ for $m = 5$.    (b) Probability that $g$ is different from Kemeny under $\mathcal{M}_\varphi^2$.

Figure 2: The ground truth $W$ and asymptotic comparisons between Kemeny and $g$ in Definition 7.

**Proof sketch:** The first part of Theorem 7 is proved by the Central Limit Theorem. For the second part, the proof for $m = 5$ uses an acyclic $W \in \mathcal{B}(\mathcal{C})$ illustrated in Figure 2 (a).    □

Theorem 6 suggests that, when $n$ is large and the votes are generated from $\mathcal{M}_\varphi^1$, it does not matter much which of $f_B^1$, $f_B^2$, and Kemeny we use. A similar observation has been made for other voting rules by Caragiannis et al. [7]. On the other hand, Theorem 7 states that when the votes are generated from $\mathcal{M}_\varphi^2$, interestingly, for some ground truth parameter, $f_B^2$ is different from the other two with non-negligible probability, and as we will see in the experiments, this probability can be quite large.

## 6.1 Experiments

We focus on the comparison between rule $f_B^2$ and Kemeny using synthetic data generated from $\mathcal{M}_\varphi^2$ given the binary relation $W$ illustrated in Figure 2 (a). By Theorem 5, the computation involves computing $\varphi^{\Omega(n)}$, which is exponentially small for large $n$ since $\varphi < 1$. Hence, we need a special data structure to handle the computation of $f_B^2$, because a straightforward implementation easily loses precision. In our experiments, we use the following approximation for $f_B^2$.

**Definition 7** *For any $c \in \mathcal{C}$ and profile $P$, let $s(c, P) = \sum_{b:w_P(b,c)>0} w_P(b,c)$. Let $g$ be the voting rule such that for any profile $P$, $g(P) = \arg\min_c s(c, P)$.*

In words, $g$ selects the alternative $c$ with the minimum total weight on the incoming edges in the WMG. By Theorem 5, the Bayesian risk is largely determined by $\varphi^{-s(c,P)}$. Therefore, $g$ is a good approximation of $f_B^2$ with reasonably large $n$. Formally, this is stated in the following theorem.

**Theorem 8** *For any $W \in \mathcal{B}(\mathcal{C})$ and any $\varphi$, $f_B^2(P_n) = g(P_n)$ a.a.s. as $n \to \infty$ and votes in $P_n$ are generated i.i.d. from $\mathcal{M}_\varphi^2$ given $W$.*

In our experiments, data are generated by $\mathcal{M}_\varphi^2$ given $W$ in Figure 2 (a) for $m = 5$, $n \in \{100, 200, \dots, 2000\}$, and $\varphi \in \{0.1, 0.5, 0.9\}$. For each setting we generate 3000 profiles, and calculate the fraction of trials in which $g$ and Kemeny are different. The results are shown in Figure 2 (b). We observe that for $\varphi = 0.1$ and $0.5$, the probability for $g(P_n) \neq \text{Kemeny}(P_n)$ is about 30% for most $n$ in our experiments; when $\varphi = 0.9$, the probability is about 10%. In light of Theorem 8, these results confirm Theorem 7. We have also conducted similar experiments for $\mathcal{M}_\varphi^1$, and found that the $g$ winner is the same as the Kemeny winner in all 10000 randomly generated profiles with $m = 5, n = 100$. This provides a check for Theorem 6.

# 7   Acknowledgments

We thank Shivani Agarwal, Craig Boutilier, Yiling Chen, Vincent Conitzer, Edith Elkind, Ariel Procaccia, and anonymous reviewers of AAAI-14 and NIPS-14 for helpful suggestions and discussions. Azari Soufiani acknowledges Siebel foundation for the scholarship in his last year of PhD studies. Parkes was supported in part by NSF grant CCF #1301976 and the SEAS TomKat fund. Xia acknowledges an RPI startup fund for support.

## Footnotes

[1]The new voting rule for $\mathcal{M}_\varphi^1$ fails them for all $\varphi < 1/\sqrt{2}$.

[2]This notation should not be taken to mean a conditional distribution over $\mathcal{S}$ unless we are taking a Bayesian point of view.

[3]In the Condorcet model the sample space is $\mathcal{B}(\mathcal{C})^n$ [31]. We study a variant with sample space $\mathcal{L}(\mathcal{C})^n$.

[4]Characterizing majority and Condorcet criterion of $f_B^1$ for $\varphi \geq \frac{1}{\sqrt{2}}$ is an open question.

[5]We require $\varphi$ to be rational to avoid representational issues.

[6]The formula resembles Young's calculation for three alternatives [34], where it was not clear whether the calculation was done for $\mathcal{F}_\varphi^2$. Recently it was clarified by Xia [31] that this is indeed the case.

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
