[Reviews · NeurIPS 2014]

Submitted by Assigned_Reviewer_9

The author(s) consider social choice mechanisms from the point of view of statistical decision theory. They use this viewpoint to derive two new voting mechanisms based on the principle of Bayesian risk minimization. The rest of the paper studies the properties of the ranking produced by the new voting mechanisms and also an in-depth study of the computational complexity of realizing these mechanisms. The decision problems arising from finding good winning candidates for one of their schemes turns out to be NP-hard while the other is in P. The paper concludes with an asymptotic comparison of the decisions produced by their mechanisms and the Kemeny voting rule and an experimental evaluation of the asymptotic results.

The results are original and the paper is well written (albeit dense with results).

Minor comments:
* Definition 4, the definition of the loss function L_{0-1}(\theta,c) is not provided this was a bit confusing since it is different from the function in Example 1. It seems that the definition should be L_{0-1}(\theta, c) = 0 if \theta \in \mathcal{L}_c(C) (resp. \mathcal{B}_c(C)) and 1 otherwise.
* Lemma 2: I found it slightly confusing that the quantities outside the statement of the lemma (lines 249-251) were used implicitly in the lemma. It would be good to clean this up so that the statement is self-contained.
* I'm not sure why this is bothering me so much, but it seems to me that the definition of P_||^NP-hard problem should use truth-table reductions rather than many-one reductions. In any case, feel free to ignore this comment since it does not affect the results of the paper.
* Typo on Page 7, line 343: "where d is not ranked" ...->... "where b is not ranked".
* Typo on Page 7, line 346: "BETTER DECISION" ...->... "OPTIMAL BAYESIAN DECISION".
Summary: The reviewer felt the author(s) have successfully treated an established area with a fresh viewpoint that also points to new and interesting open questions.

Submitted by Assigned_Reviewer_13

The paper casts the problem of designing voting rules (choosing an order over a set of alternatives) for multiple agents as decisions that minimize the expected loss w.r.t the posterior distribution over parameters i.e. Bayesian risk.

To the best of my assessment, the results are technical correct, novel and interesting.

My only comment or concern, is that the paper presents normative properties and asymptotic results (as number of voters grow large) of the Bayesian estimators. The NIPS audience tends more interested in learning dynamics. Given that I perceive the results to be of high-quality, I don't know if NIPS is the best match for this paper.
Summary: The paper presents the problem of designing voting rules (choosing an order over a set of alternatives) for multiple agents as minimizing the expected loss w.r.t the posterior distribution over parameters i.e. Bayesian risk. The normative and asymptotic results are novel and interesting.

Submitted by Assigned_Reviewer_35

Recall that there are tons of different voting rules. How can single any out, or propose new ones, in a principled way? One interesting line of work in computational social choice (e.g., Conitzer-Sandholm) asks which rules arise as the MLE in some setting (some do, some don't). This submission proposes reversing the process: posit a statistical decision-making model (prior over a ground truth and a loss function), solve for the optimal (i.e., expected loss-minimizing) rule, and see if you get something interesting.

The authors carry out the general plan above in two variants of the Mallows model. In both cases there is a ground truth, and intuitively each pairwise ordering is flipped independently according to some noise parameter. The two models differ as to whether the ground truth is required to be consistent (i.e., a linear ordering) or not. The rules produced by the above paradigm are not necessarily the same as any of the well-studied rules. The authors study the new rules along three dimensions: first, which of the standard properties do they possess (e.g., monotonicity); second, in the limit with n iid inputs, how does their output compare to the well-known Kemeny rule; third, the computational complexity of computing these rules (hard when ground truth is a linear order, easy otherwise).

While none of the individual results are earth-shattering, the high-level idea is nice and the collection results is a thorough study of one natural instantiation of the framework.
Summary: While none of the individual results are earth-shattering, the high-level idea is nice and the collection results is a thorough study of one natural instantiation of the framework.
Author Feedback
Author rebuttal: We thank all reviewers for useful comments!

To Reviewer 1: We think that our proposed statistical approach towards social choice is relevant due to the strong interdisciplinary nature of NIPS, which was emphasized in the call for papers. We also feel that this new direction is interesting to and will benefit from the statistical and machine learning audience attending NIPS.

We will follow reviewer 3's suggestion to clean up the presentation. We will also put a full version online once the paper is accepted.